# Variants in *ASPH* cause exertional heat illness and are associated with malignant hyperthermia susceptibility

Yukari Endo [1], Linda Groom[2], Alper Celik[3], Natalia Kraeva[4], Chang Seok Lee[5], Sung Yun Jung[5], Lois Gardner[6], Marie-Anne Shaw[6], Susan L. Hamilton[5], Philip M. Hopkins [6,7], Robert T. Dirksen[2], Sheila Riazi [4✉] & James J. Dowling [1,8,9,10✉]

Exertional heat illness (EHI) and malignant hyperthermia (MH) are life threatening conditions associated with muscle breakdown in the setting of triggering factors including volatile anesthetics, exercise, and high environmental temperature. To identify new genetic variants that predispose to EHI and/or MH, we performed genomic sequencing on a cohort with EHI/MH and/or abnormal caffeine-halothane contracture test. In five individuals, we identified rare, pathogenic heterozygous variants in *ASPH*, a gene encoding junctin, a regulator of excitation-contraction coupling. We validated the pathogenicity of these variants using orthogonal pre-clinical models, CRISPR-edited C2C12 myotubes and transgenic zebrafish. In total, we demonstrate that *ASPH* variants represent a new cause of EHI and MH susceptibility.

[1] Program for Genetics and Genome Biology, Hospital for Sick Children, Toronto, Ontario, Canada. [2] Department of Physiology, University of Rochester, Rochester, NY, USA. [3] Centre for Computation Medicine, Hospital for Sick Children, Toronto, Ontario, Canada. [4] Malignant Hyperthermia Unit, Department of Anesthesia, Toronto General Hospital, Toronto, Ontario, Canada. [5] Department of Molecular Physiology and Biophysics, Baylor College of Medicine, Houston, TX, USA. [6] Leeds Institute of Medical Research at St. James's, University of Leeds, Leeds, UK. [7] Malignant Hyperthermia Unit, St. James's University Hospital, Leeds, UK. [8] Division of Neurology, Hospital for Sick Children, Toronto, Ontario, Canada. [9] Department of Paediatrics, University of Toronto, Toronto, Ontario, Canada. [10] Department of Molecular Genetics, University of Toronto, Toronto, Ontario, Canada. ✉email: Sheila.riazi@uhn.ca; james.dowling@sickkids.ca

Exertional heat illness (EHI) comprises a spectrum of conditions characterized by core body hyperthermia and muscle breakdown that can lead to multi-organ failure and ultimately death[1]. EHI represents a leading cause of mortality among young athletes, and is a feared complication in settings such as basic military training and intense athletic performance[2–5]. Any factor that reduces body heat dissipation, such as high environmental temperature, lack of acclimatization, or dehydration, can increase the risk of EHI; however, a subset of individuals that develop the most severe forms of EHI have a genetic predisposition. EHI, along with malignant hyperthermia (MH) and recurrent rhabdomyolysis[6,7], represent dynamic muscle conditions that are thought to share common pathomechanisms[8,9].

There are currently no primary treatments for EHI, and prompt supportive care during an event is required to prevent systemic injury and death[10]. Because of this, there is a critical need to identify individuals pre-disposed to developing EHI. Predisposition is thought to have a strong genetic underpinning, though to date only variants in the gene that encodes the type I skeletal muscle ryanodine receptor (RYR1) have been convincingly associated with EHI[11]. Because of likely shared pathophysiology, genes associated with MH are considered candidates for EHI. The genetics of MH susceptibility (MHS) has been more thoroughly studied. It is estimated that approximately 70% of the genetic burden of MHS is identified, with more than 90% of genetically solved individuals harboring a pathogenic variant in RYR1, and the remainder having variants in CACNA1S or STAC3[9,12]. RYR1 variants are also the most common cause of recurrent rhabdomyolysis[13]. Importantly, however, the genetic cause of the additional ~30% of MH susceptibility remains unknown, and a much larger percentage of EHI genetic burden unresolved.

RYR1 encodes the core component (RyR1) of a larger multi-protein calcium release complex (CRC)[14,15]. This complex resides within the skeletal muscle at the triad, a structure that represents the apposition of the transverse tubule (T-tubule) and the terminal cisternae of the sarcoplasmic reticulum[16]. It governs the process of excitation-contraction coupling (ECC), wherein a nerve stimulus is converted from membrane depolarization to muscle contraction via the regulated release of calcium through RyR1. Dynamic muscle diseases such as EHI and MHS are thought to result from uncontrolled RyR1-mediated calcium release at the triad in response to triggering exposures such as volatile anesthetics, heat, and exercise[17–19].

The overall goal of this study is to identify new genetic causes of EHI/MHS. The genes encoding the multi-protein CRC are all considered potential candidates for EHI/MHS. One such candidate is junctin, encoded by the ASPH (Aspartate Beta-Hydroxylase) gene[20,21]. Junctin is localized to the lumen of the terminal sarcoplasmic reticulum (SR), where it has been shown to interact with calsequestrin and RyR1 and participate in the regulation of SR calcium dynamics[22]. In this study, we identify pathogenic ASPH variants in five individuals with EHI/MHS. We additionally show that these variants cause phenotypes in C2C12 cells and transgenic zebrafish that mirror the human disease and thus support their pathogenicity. In summary, we conclude that junctin variants represent a new genetic cause of EHI/MHS.

## Results

### Genome-scale sequencing of a cohort of individuals with MHS/EHI.
The goal of this study was to identify new genetic causes of MHS and EHI. To accomplish this, we performed genome-scale sequencing on a cohort of individuals referred to the Toronto General Hospital Malignant Hyperthermia Unit. The cohort is composed of individuals who either have had a sentinel event of EHI or MH, or else have a positive caffeine-halothane contracture test (CHCT) and a first degree relative with EHI/MH. The CHCT is a gold standard, biopsy-based test for MHS with a caffeine and halothane challenge component[23]. All affected individuals (history of EHI/MH and/or positive CHCT) were screened for variants in RYR1 and CACNA1S by Sanger sequencing, and sequence negative patients were carried forward to the mutation discovery arm of the study.

103 individuals (63 affected individuals from 34 families plus 40 additional sporadic cases) were studied by whole-exome or whole-genome sequencing. Based on the known pathomechanisms underlying EHI/MH, variants in genes encoding components of the excitation-contraction coupling machinery are strong causal candidates. Therefore, after sequence alignment and variant filtration, we focused analyses on a list of 22 candidate genes associated with ECC (Supplementary Table 1), with prioritization of variants based on a minor allele frequency (MAF) of <0.01 in The Genome Aggregation Database (gnomAD)[24]. Using this strategy, 48 rare variants were identified (Supplementary Table 2), of which 35 were in RYR1 or CACNA1S. As described in detail below (Supplementary Table 3), three patients had variants in the junctin isoform of the ASPH gene[25].

### Identification of junctin variants in individuals with EHI.
Two affected individuals from one family were found to have a heterozygous missense variant in the junctin transcript of ASPH (ENST00000522603.1: c.161T > C; ENSP00000436188.1: p.Val54Ala) (Table 1 and Fig. 1a). Both presented with myalgias exacerbated by heat and/or exercise, and one had documented elevation in serum creatine kinase (CK) with exercise. Both had positive CHCT for halothane and caffeine (MHS). An unaffected sibling (negative CHCT = MH negative) did not carry the variant. A third unrelated affected individual with a confirmed episode of MH plus a positive CHCT was found to harbor a heterozygous ASPH variant (c.445G>C, p.Asp149His) (Table 1 and Fig. 1a). In addition to MH, he had non-anesthetic symptoms of myalgias and muscle cramps worsened by heat and exercise.

We next screened a UK cohort of individuals with exertional heat illness for additional ASPH variants. Two individuals were identified with heterozygous ASPH variants (c.263A > C, p.Lys88Thr and c.605A > G, p.Lys202Arg) from a cohort of 64 EHI patients examined with a targeted next-generation sequencing panel[26] (Table 1 and Fig. 1a). Both patients experienced EHI episodes and were diagnosed as MHN by in vitro contracture test (IVCT) (Table 1)[27].

In total, 5 individuals from a total of 167 cases were found to have rare, potentially damaging heterozygous missense variants in ASPH. All variants were located within the junction isoform of the ASPH gene (Supplementary Fig. 1). Junctin expression is restricted to skeletal and cardiac muscle, with skeletal muscle junctin encoded by 5 exons and cardiac junctin by 6 exons. For the skeletal muscle isoform, exon 1 encodes the cytoplasmic portion, while exons 2–5 encode the transmembrane and luminal portions of the protein (Supplementary Fig. 2)[28]. All 4 variants that we identified encode amino acids in the luminal domain (Fig. 1b). All are predicted to be deleterious by at least one prediction modeling program (Table 1).

### Validation of junctin variant pathogenicity using the zebrafish model system.
We took two approaches for validating the pathogenicity of the identified junctin variants. The first was to express the variants in zebrafish and assess the resulting muscle phenotype(s). To accomplish this, we created transgenic zebrafish

**Table 1 Identification of rare, pathogenic *ASPH* variants in four families with EHI/MHS.**

| Variant name | V54A | K88T | D149H | K202R |
|---|---|---|---|---|
| Chr | 8 | 8 | 8 | 8 |
| Position (GRCh 37) | 62596603 | 62580807 | 62577998 | 62577838 |
| Ref | A | T | C | T |
| Allele | G | G | G | C |
| Consequence | missense_variant | missense_variant | missense_variant | missense_variant |
| SYMBOL | *ASPH* | *ASPH* | *ASPH* | *ASPH* |
| CDS_position | 161 | 263 | 445 | 605 |
| Exon | 2 | 4 | 5 | 5 |
| Protein_position | 54 | 88 | 149 | 202 |
| Amino_acids | V/A | K/T | D/H | K/R |
| Codons | gTt/gCt | aAa/aCa | Gac/Cac | aAg/aGg |
| Existing_variation | rs779765042 | – | rs1236122813 | rs145678786, COSV100727527 |
| gnomAD_AF | 0.0001076 | – | 0.00001924 | 0.0008436 |
| SIFT | deleterious(0.01) | tolerated(0.07) | deleterious(0.04) | deleterious_low_confidence(0) |
| PolyPhen | probably_damaging(0.996) | probably_damaging(0.976) | possibly_damaging(0.824) | benign(0.084) |
| CADD_PHRED | 27.9 | 22.5 | 16.1 | 9.921 |
| Clinical diagnosis | MHS/EHI | EHS | MHS/EHI | EHS |
| Contracture test | CHCT | IVCT | CHCT | IVCT |
| Caffeine | 0.4*(proband), 0.8*(sister) | 0 g | 0.6* | 0 g |
| Halothane | 1.4**(proband), 1.0**(sister) | 0.05 g | 4.0** | 0.1 g |

The threshold responses for a positive diagnosis in CHCT were *caffeine >0.3 and/or **halothane >0.7. The threshold for a positive diagnosis in IVCT were 0.2 g force produced at 2 mM caffeine and/or at 2%.

expressing human junctin cDNA fused to mCherry in a muscle-specific manner using the 503unc promoter[29]. The junctin cDNAs contained either wild-type sequence or the V54A or K88T variants identified in our patient cohort. Three independent lines were created per transgene, and all analyses were done on F2 generations or later. We first confirmed transgene mRNA levels using qRT-PCR (Supplementary Fig. 3) and protein expression and localization to the triad using immunofluorescent microscopy (Fig. 2a). Of note, WT and K88T junctin mRNA levels were comparable, while expression of V54A mRNA was reduced as compared to the other lines.

**Transgenic mutant junctin zebrafish have abnormal responses to heat and caffeine**. We performed general characterization of the transgenic zebrafish lines, and identified no obvious differences in baseline motor behavior (Fig. 2d and Supplementary Fig. 4a) and muscle ultrastructure (Fig. 2b). This was expected given that patients with EHI/MHS do not typically exhibit overt signs of disease outside of their acute episodes. To determine if junctin mutant transgenic fish exhibit a dynamic phenotype similar to EHI/MHS, we developed a novel assay whereby we challenged embryonic fish with heat (34 °C) plus 1 mM caffeine and then measured motor activity using an automated movement tracking system (Fig. 2c). Non-transgenic fish and fish expressing WT junctin had normal/unchanged swim behavior with this exposure paradigm. Conversely, both V54A and K88T transgenic fish exhibited significantly reduced swim parameters after heat and caffeine treatment (Fig. 2d and Supplementary Fig. 4a).

To determine if all identified junctin variants impact motor function, we injected junctin mRNA harboring different variants into 1 cell stage embryos, and then performed the heat (34 °C) plus 1 mM caffeine exposure assay. Similar to the transgenics, fish injected with junctin mRNA containing any of the 4 patient variants exhibited significantly reduced distance traveled on swim analysis when exposed to heat and caffeine (Fig. 2e, right panel and Supplementary Fig. 4b). In addition, as compared to WT junctin expressing embryos, fish expressing V54A or K88T

mutant junctin mRNAs also exhibited reduced swimming with heat exposure only.

One potential mechanism for EHI/MHS, as demonstrated for *RYR1* variants, is altered RyR1 dependent calcium release[17,18,30]. We examined this possibility in the context of junctin variants using myofibers freshly isolated from transgenic fish. Interestingly, a leftward shift in the caffeine concentration curve in K88T transgenic myofibers was noted (Fig. 3a). This type of change is consistent with increased RyR1-mediated calcium release at lower levels of stimulation, consistent with increased sensitization of RyR1 (a hallmark of MH-associated mutations).

MH-related *RYR1* variants are also associated with increased oxidative stress[31]. We measured reactive oxygen species (ROS) in junctin transgenic zebrafish using the DCFDA assay. No change in ROS production was seen in non-exposed transgenics. Heat and caffeine exposed junctin K88T transgenics, however, exhibited significantly increased ROS production (Fig. 3b). Interestingly, another activator of RyR1, 4-Chloro-3-methylphenol (4-CMC), when combined at 10 µM concentration with heat, also increased ROS production in both K88T and V54A transgenic zebrafish (Fig. 3b).

**Transgenic mutant junctin zebrafish undergo heat plus caffeine provoked myofiber damage that is prevented by dantrolene or N-acetylcysteine treatment**. Dantrolene is an effective pharmacotherapy for MH[12], and is also useful for reducing muscle cramping in patients that experience non-anesthetic MH-like events. In addition, the anti-oxidant N-acetylcysteine (NAC) reduces oxidative stress and prevents adverse responses to heat in a mouse MH model with *Ryr1* mutation[31]. Therefore, we tested whether treatment with either dantrolene or NAC could prevent the defective swim behavior in K88T junctin transgenic zebrafish during the heat and caffeine exposure paradigm detailed in Fig. 2c. We found that both 5 µM dantrolene and 200 µM NAC treatment prevented the reduction in swim behavior observed in K88T transgenic fish, restoring total distance traveled to wild-type levels (Fig. 3c and Supplementary Fig. 5).

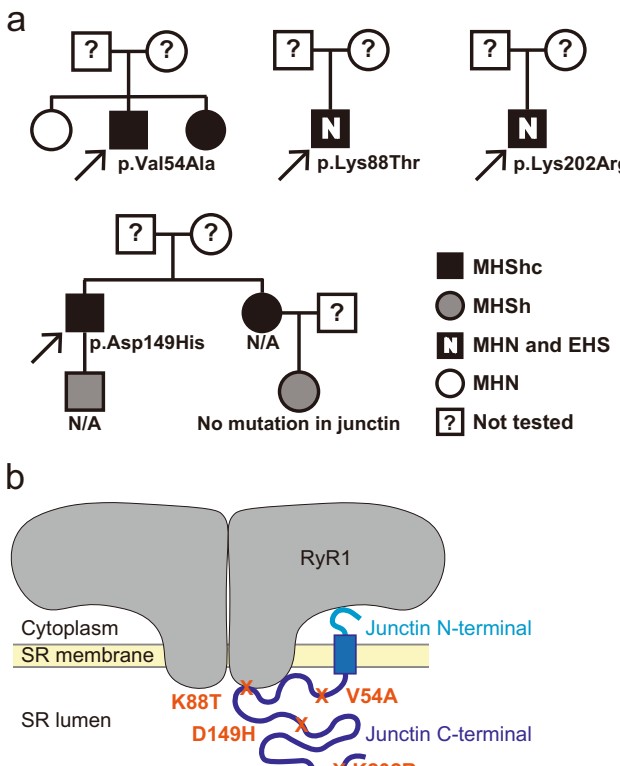

**Fig. 1 Identification of heterozygous, missense junctin variants in four families with EHI/MHS. a** Pedigrees of four families with exertional heat illness (EHI) and/or malignant hyperthermia susceptibility (MHS) and with rare junctin variants. Caffeine halothane contracture test (CHCT) results are listed as follows: MHShc = MH susceptible to both halothane and caffeine; MHSh = MH susceptible to halothane alone; MHN = CHCT test negative. N/A = DNA not available for genetic evaluation. Arrow = proband. **b** Positions of the four junctin variants are shown within the protein domain structures of junctin. The binding sites for RyR1 are also depicted.

Muscle breakdown is an important sequalae associated with episodes of EHI/MH[9]. Thus, we examined whether muscle integrity was impacted in our junctin transgenic zebrafish with heat/caffeine exposure. We measured fiber integrity by performing whole-mount immunostaining staining with an anti-myosin heavy chain antibody and scoring muscle with a severity scale that incorporates separation of myofibers from the myosepta (Fig. 3d, e). Under normal conditions, no muscle breakdown was observed in wild type or K88T transgenic zebrafish embryos. With heat + caffeine exposure, however, occasional detached fibers were seen in WT transgenics, which was partially prevented with dantrolene (but not NAC) pre-treatment. Conversely, widespread loss of myofiber integrity was seen in K88T transgenic fish following heat + caffeine exposure, which was also partially prevented by either dantrolene or NAC treatment. Taken together, these data show that variants in junctin, when expressed in zebrafish, impact muscle structure and function in the setting of heat + caffeine, which strongly supports the pathogenicity of these variants.

**C2C12 cells with junctin variants have abnormal calcium dynamics and aberrant oxidative stress.** To corroborate our findings in zebrafish in a mammalian context, we generated, by CRISPR gene editing, knock-ins (KI) mirroring patient variants in the skeletal myocyte C2C12 cell line. We created heterozygous

KI of V54A and homozygous KI of K88T, along with another line containing homozygous frameshift/stop (Supplementary Fig. 6). We examined expression by western blot, and found reduced junctin levels in K88T C2C12 cells and no detectable junctin protein in the frameshift line (Fig. 4a, Supplementary Fig. 7). While no differences in caffeine sensitivity were observed at either room temperature or 37 °C (Fig. 4b), or in resting calcium levels at room temperature, a large increase in resting calcium was observed in all mutant lines with exposure to heat (RT for 45 min and then heated at 37 °C for 10 min) and 50 μM 4-CmC (Fig. 4c). These data are consistent with excessive RyR1 calcium leak under these conditions in the setting of junctin variants.

We next assessed oxidative stress levels using the same DCFDA assay employed in zebrafish (Fig. 4d, Supplementary Fig. 8). Small, but significant, increases in ROS production were observed with heat challenge (RT for 1 h and then 37 °C for 30 min), in cells harboring either the junctin K88T variant or the frameshift mutation. No changes were seen with heat plus caffeine exposure, but significant (and larger magnitude) changes were seen in all 3 lines with heat plus 50 μM 4-CmC.

**Junctin variants impact RyR1 phosphorylation and CASQ1 expression.** Junctin binds to RyR1 and modulates its function[21,32–36]. Using RyR1 immunoprecipitation followed by mass spectrometry with parallel reaction monitoring to assess phosphorylation, we found that RyR1 immunoprecipitated from C2C12 cells was phosphorylated at residue S2844 (a major RyR1 phosphorylation site) at a level of $11.0 \pm 2.5\%$ ($n = 4$). When compared to these controls, the S2844 phosphorylation of immunoprecipitated RyR1 from K88T junctin C2C12 cells was significantly upregulated (19%, Fig. 4e). While increased RyR1 phosphorylation at S2844 is associated with increased RyR1 $Ca^{2+}$ leak[37], we are not able to fully conclude that the phosphorylation changes seen in K88T variant C2C12 cells are responsible for the increased calcium sensitivity of RyR1 to heat and 4-CmC (Fig. 4c) observed in those cells.

Junctin also interacts calsequestrin-1 (CASQ1)[38], a key regulator of SR calcium homeostasis[39]. We examined CASQ1 levels in our modified C2C12 cells, and found reduced expression (as compared to wild type) in all three variant lines (Fig. 4a, Supplementary Fig. 7b). To determine if this may be associated with reduced junctin-CASQ1 association, we returned to our transgenic zebrafish lines, and performed AP-MS using anti-mCherry immuno-capture. As compared to mCherry-WT-junctin, mCherry-K88T-junctin had reduced association with CASQ1 (Fig. 4f). We also identified additional junctin interactors, several of which (particularly SERCAs and mitochondrial complex V ATPases) are differentially bound to wild type vs mutant junctins (Supplementary Figs. 9 and 10). Of note, we could not identify an anti-CASQ1 antibody that works in zebrafish, so could not examine CASQ1 localization or expression levels in our transgenic models.

**Discussion**

In this study, we identified rare, pathogenic variants in junction (*ASPH*) as a novel genetic cause of exertional heat illness and malignant hyperthermia susceptibility. We validated the pathogenicity of these variants using two orthogonal pre-clinical model systems, zebrafish and CRISPR-edited C2C12 myotubes, and in doing so establish new experimental paradigms for modeling EHI and MHS. In these models, we further showed that treatment with dantrolene or N-acetylcysteine can reduce changes provoked by heat exposure, suggesting potential treatment avenues for individuals with junctin-associated EHI/MHS. Given the critical importance of detection and prevention of EHI/MH, we advocate

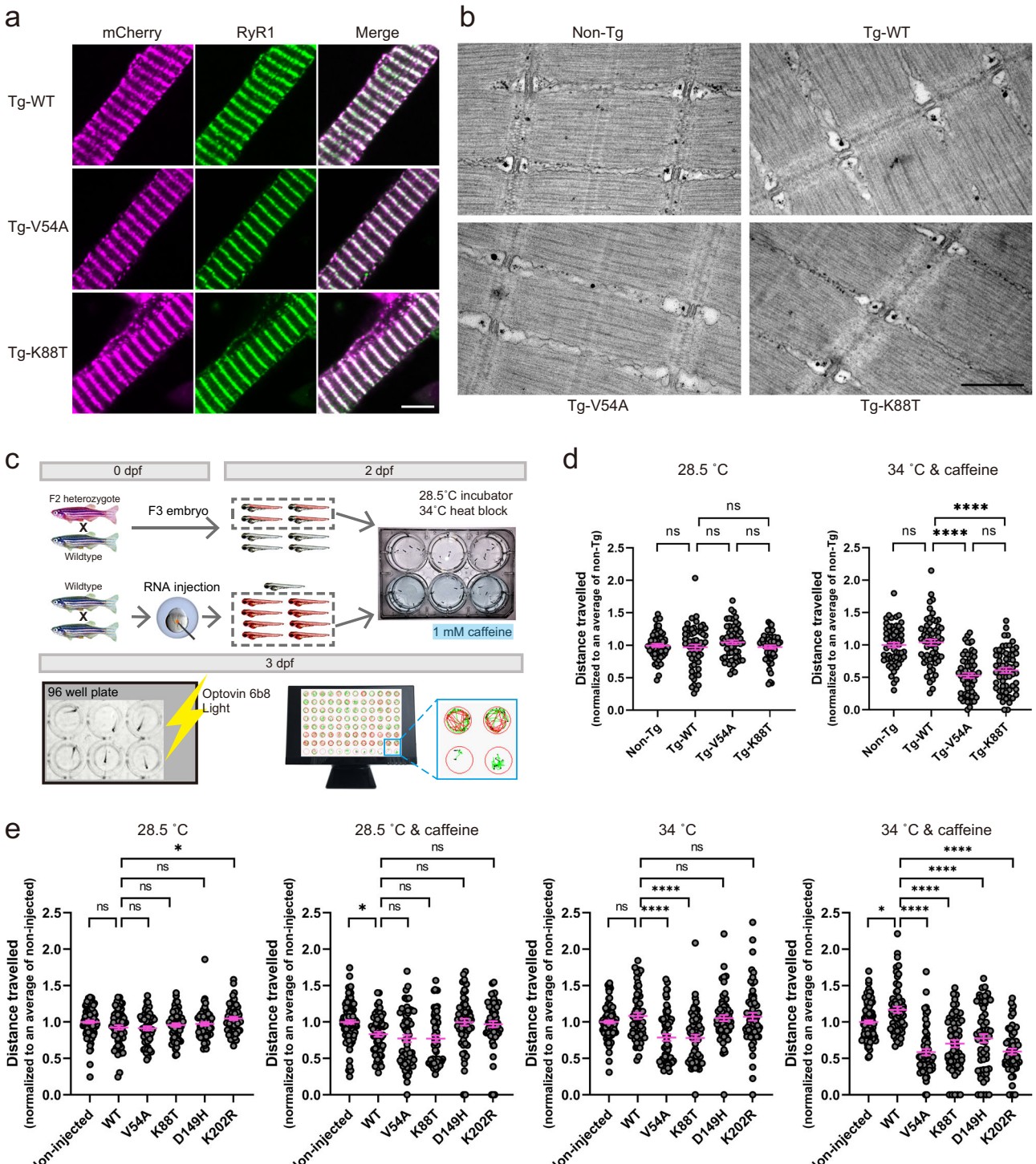

**Fig. 2 Junctin variant transgenic zebrafish exhibit heat- and caffeine-induced motor defects. a** Immunolabeling of myofibers isolated from transgenic (Tg)-WT, Tg-V54A and Tg-K88T zebrafish with anti-mCherry and anti-RyR1antibodies. Magenta: mCherry, green: RyR1, scale bar: 5 μm. **b** Representative electron microscopy images from muscle of non-Tg, Tg-WT, Tg-V54A, and Tg-K88T zebrafish. Scale bar: 500 nm. **c** An overview of the workflow of the Heat + Caffeine assay. **d** Swimming ability of junctin mutant transgenic fish is altered with heat + caffeine exposure. Fish were tested without treatment (left panel) or with 1 mM caffeine at 34 °C for 1 h (right panel). Distance traveled was measured and normalized to the mean of non-Tg tested from the same day. Each group consists of 60 fish, which were obtained from three independent experiments with 20 fish per group. Error bars represent SEM. **e** Synergistic effect of heat and caffeine on the swimming ability of fish injected with junctin mRNA. Scatter plots display the total distance traveled normalized to the mean of non-injected measured on the same day. Data from non-injected ($n = 100$) and the other groups ($n = 60$) were from independent experiments with 20 animals per group at a time. The treatment conditions for each are no treatment (the leftmost panel), 1 mM caffeine at 28.5 °C for 1 h (the second panel from the left), at 34 °C for 1 h (the second panel from the right), and 1 mM caffeine at 34 °C for 1 h (the rightmost panel). Error bars represent SEM. Statistical analysis for (**d**, **e**) by one-way ANOVA followed by Tukey's (**d**) and Dunnett's (**e**) multiple comparisons tests where $*p < 0.05$, $***p < 0.001$, $****p < 0.0001$. Source data are provided as a Source data file.

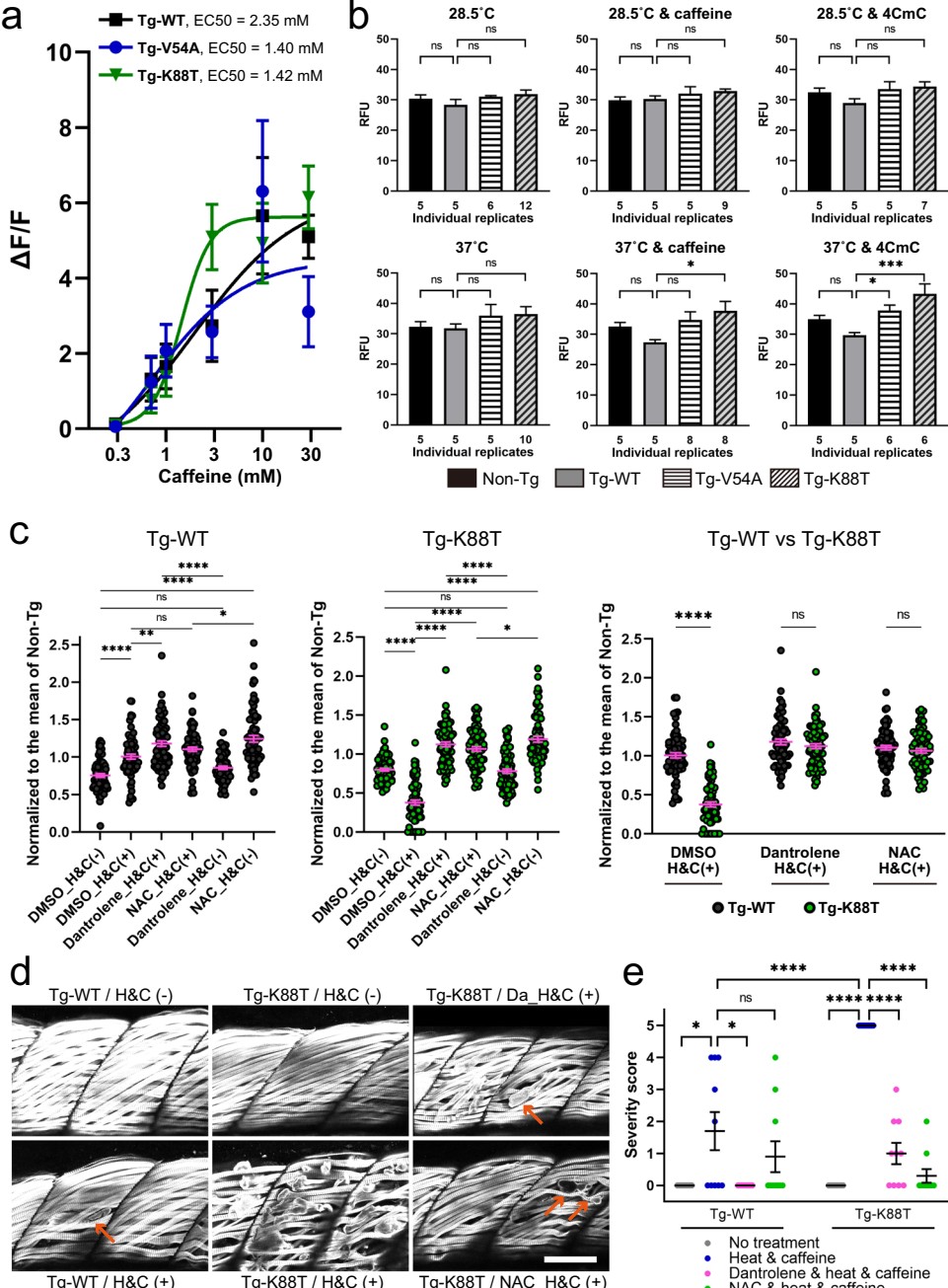

**Fig. 3 Junctin mutant zebrafish exhibit heat plus caffeine provoked cellular pathology that is rescued with dantrolene or N-acetylcysteine.**
**a** Concentration dependence of caffeine-induced calcium release in Tg-fish myofibers at room temperature. A leftward shift is seen with K88T (see also Supplementary Table 4 for the numbers of myotubes analyzed). **b** 2′,7′-dichlorofluorescin diacetate (DCFDA) assay of reactive oxidative species (ROS) reveals increased ROS production with heat plus caffeine or 4-CmC in V54A and K88T transgenic fish. ROS production in 6-dpf fish was measured under the following conditions; (1) no treatment at 28.5 °C, (2) with 1 mM caffeine at 28.5 °C, (3) with 10 μM 4CmC at 28.5 °C, (4) no drug at 37 °C, (5) with 1 mM caffeine at 37 °C, and (6) with 10 μM 4CmC at 37 °C. A replicate consists of 20 fish. Numbers of independent replicates are shown below each bar. Statistical analysis was one-way ANOVA followed by Dunnett's multiple comparisons test. **c–e** Dantrolene (Da) and N-acetylcysteine (NAC) protect against heat + caffeine (H&C) induced injury in Tg-WT and Tg-K88T fish. **c** Swim assay using methodology from Fig. 2c, with 72 fish in each group (three independent experiments with $n = 24$ fish). Swimming distance of each fish was normalized to the mean of non-Tg with the same condition from the same day. Statistical analysis by one-way ANOVA followed by Tukey's multiple comparisons test. **d** Whole-mount immunostaining of transgenic fish after heat + caffeine challenge. Anti-myosin antibody staining was used to illuminate myofibers. High incidence of severe myofiber detachment (arrows) was observed in Tg-K88T exposed to H & C, which was reduced with dantrolene or NAC treatment. Scale bar: 50 μm. **e** Quantification of muscle damage after a heat and caffeine challenge. The severity of the damage was scored as described in Methods and presented as scattered plots. Ten fish per group were assessed, and the difference between group means was analyzed by Mann–Whitney U test. All data are represented as mean ± SEM. Differences were considered to be statistically significant at $P < 0.05$ (*), $P < 0.01$ (**), $P < 0.001$ (***) or $P < 0.0001$(****). Source data are provided as a Source data file.

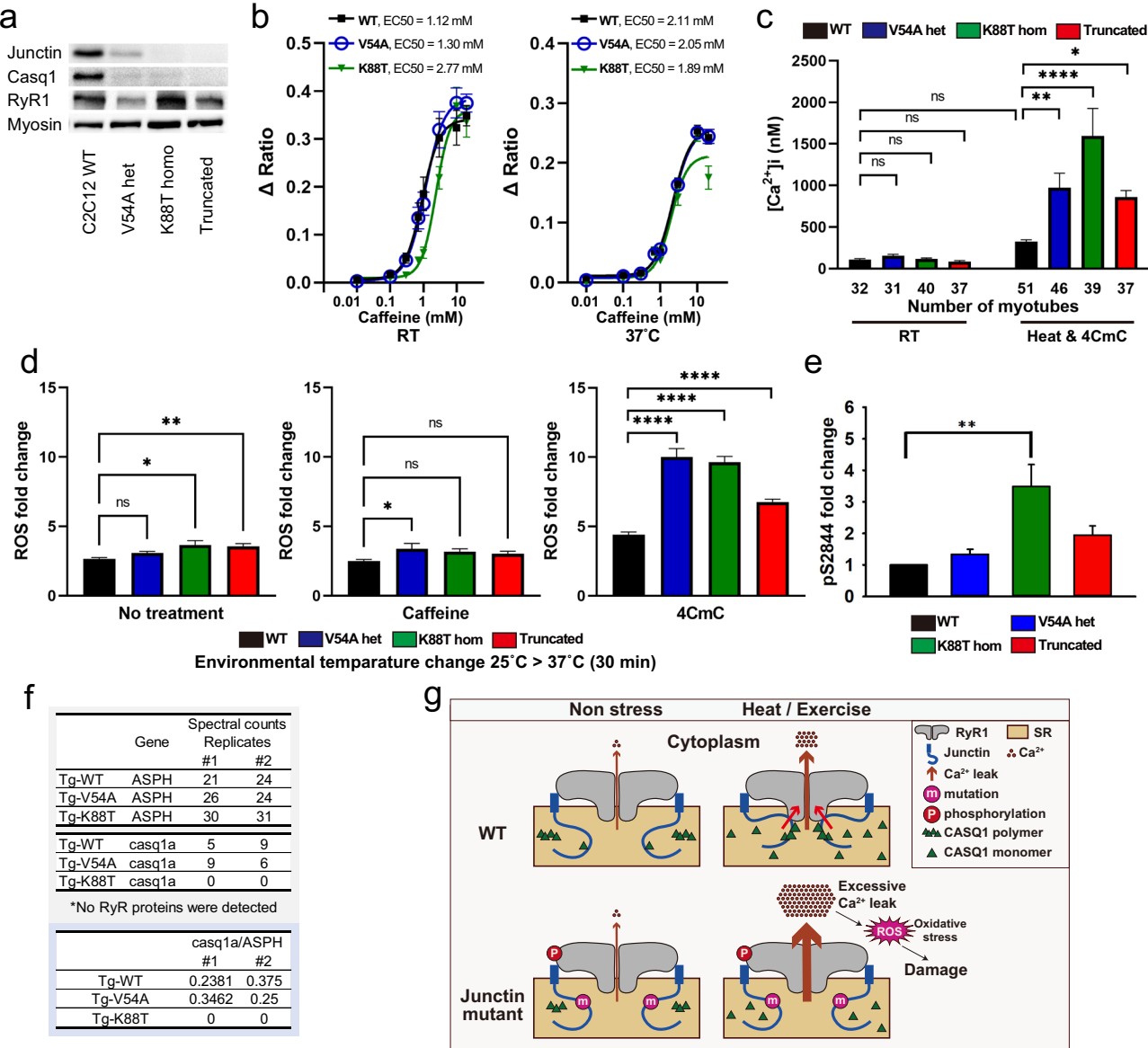

**Fig. 4 Analysis of junctin variant knock-in C2C12 myotubes. a** Junctin expression in C2C12 myotubes was assessed by western blotting. Myosin heavy chain was used as a loading control and a differentiation marker and was probed on the same blot as junctin. Blots for Casq1 and RyR1 were processed in parallel using the samples derived from the same experiment. Total protein was also used as loading controls (Source data file). **b** Concentration dependence of caffeine-induced SR calcium release in C2C12 myotubes. Left panel; at room temperature, 4 replicates in each cell line. Right panel; at 37 °C, 8 replicates in WT and V54A, 9 replicates in K88T. **c** Resting intracellular calcium concentrations in C2C12 myotubes. Cytosolic calcium concentration was measured at RT or after heat and 50 μM 4CmC exposure. The numbers of myotubes analyzed are shown below each bar. Statistical analysis was by one-way ANOVA followed by Dunnett's multiple comparisons test. **d** DCFDA assay in C2C12 shows increased temperature-dependent ROS production in junctin mutant cell lines during 50 μM 4CmC exposure. ROS was measured from 6-day differentiated myotubes after changes in environmental temperature (25 °C for 1 h followed by 37 °C for 30 min) with either no treatment (left panel), 1 mM caffeine (middle panel), or 50 μM 4CmC (right panel). Statistical analysis of three independent replicates was done using one-way ANOVA followed by Dunnett's multiple comparisons test. **e** A quantification of RyR1 phosphorylation (relative to WT) as measured by mass spectrometry and confirmed by Western blot analyses. Increased S2844 phosphorylation was seen in K88T cells as compared to controls. **f** Spectral counts of junctin and Casq1 proteins analyzed by AP-MS from junctin transgenic zebrafish ($n = 2$ biological replicates). All data in (**b**–**e**) are represented as mean ± SEM. Differences were considered to be statistically significant at $P < 0.05$ (*), $P < 0.01$ (**), $P < 0.001$ (***) or $P < 0.0001$(****). Source data are provided as a Source data file. **g** Schematic summarizing the results of the study and the predicted impact of junctin variants on CASQ1, RyR1, and stress-related calcium leak.

for adding *ASPH*, and specifically the variants we identified, to the genetic screening of individuals who experience anesthesia or heat-related episodes of muscle injury, and consideration of pathogenic junctin variants as reportable secondary findings from genome level testing performed for other indications.

Our data expand the knowledge of genes with causative variants that are associated with EHI/MH, adding to a list that

includes *RYR1*, *CACNA1S*, and *STAC3*. A large body of literature exists to support pathogenicity of *RYR1* variants in MH (reviewed in ref. [40]), whereas only a few variants in *CACNA1S* and *STAC3* are validated[41,42]. Our results most strongly support inclusion of the V54A and K88T variants in junctin as validated causes of EHI/MHS, with the V54A variant also associated with MH based on observation of a bona fide MH reaction in one individual and

positive CHCT in his sibling. The data are also consistent with pathogenicity of D149H and K202R for EHI, while D149H was also associated with MHS (based on CHCT). Whether other *ASPH* variants can cause EHI/MHS remains to be seen. They will likely require validation, either through orthogonal approaches such as those used in this study, and/or through repeated observation in multiple unrelated individuals.

Prediction of which variants may be pathogenic may be inferred from the limited knowledge of junctin functional domains. For example, KEKE motifs in the luminal aspect of junctin mediate interaction with calsequestrin and regulate calcium homeostasis[33,34,43]. In fact, the K88T variant identified and validated in our study is located within a KEKE motif; thus, we predict that variants found in this and other KEKE motifs are likely to confer risk of EHI. In support of this, there are very few variants in gnomAD in KEKE motifs. The functional role(s) of other junctin motifs/domains are less well understood. Rare variation is reported in gnomAD for many junctin encoded amino acids residues outside of the KEKE motifs, thus making it critical to define which changes are likely pathogenic and which are not. An approach that combines variant scanning[44] with functional validation, if a suitable high throughput assay can be established, is an attractive strategy for evaluating the potential pathogenicity of other junctin variants.

The root cause of EHI/MH is calcium dysregulation in response to a triggering exposure, be it volatile anesthetic or heat plus exercise. Ample data in the literature support a primary role for RyR1 hyper-excitability in this process[45], as evidenced by in vitro modeling of *RYR1* variants associated with MHS and study of knock-in mice with *Ryr1* MHS mutations[46]. Junctin is a known but complex regulator of RyR1 function[47]. By binding to both RyR1 and calsequestrin, junctin is proposed to mediate an interplay between these two calcium handling proteins, and thereby regulate the impact of calcium in the lumen of the sarcoplasmic reticulum on RyR1 activity. Our demonstration of increased heat-induced oxidative stress and resting calcium in junctin mutant models is consistent with these functions (Fig. 4g).

One attractive hypothesis for how junctin variants predispose to EHI and MHS is that junctin regulates SR luminal calcium homeostasis by modulating the levels and/or localization of calsequestrin, a key SR calcium binding protein. This is consistent with our findings that calsequestrin expression is decreased in junctin variant C2C12 cells, and that calsequestrin binding to mutant K88T junctin is reduced in transgenic zebrafish. Our data also are in line with the observed phenotypes of *Casq1* knockout mice[48,49]. Another potential means via which mutant junctin may act is through modulation of RyR1 activity, which is regulated in part by RyR1 phosphorylation status. The observation of increased RyR1 S2864 phosphorylation in mutant C2C12 cells is consistent with this possibility. Future research will be required to test these ideas and to dissect the specific role(s) of abnormal junctin in EHI/MH.

Standard care for MH reactions involves dantrolene administration[50]. Dantrolene binds RyR1 and reduces ECC and SR calcium release. We found dantrolene treatment to be effective in ameliorating changes in both our zebrafish and C2C12 cell junctin models, supporting the causal relationship between junctin variants and MH/EHI, and further suggesting that these variants promote MH/EHI via aberrant RyR1 calcium release. Based on these data, it is likely that patients with junctin variants who develop MH will respond to dantrolene. Our pre-clinical findings also support consideration of dantrolene as a potential therapy for junctin-related EHI, though this strategy has not met with much success in the general treatment of EHI. More chronic symptoms, such as myalgias or muscle cramps with exercise, may be more responsive. Our finding that N-acetylcysteine also can ameliorate aspects of the pre-clinical phenotypes is consistent with data from *RYR1* EHI/MHS models, confirming aberrant oxidative stress as an important aspect of the pathobiology of these conditions. The role of anti-oxidants as a therapy in EHI/MHS is still to be fully studied; a recent clinical trial of N-acetylcysteine in patients with *RYR1* related myopathy identified aberrant oxidative stress but treatment failed to show improvement in motor phenotypes or reduction of ROS[51].

In summary, we identified and validated a new genetic cause of exertional heat illness, a finding that has important implications in terms of primary prevention and subsequent clinical management for this devastating and important condition. We also establish novel experimental strategies for testing variants associated with EHI/MHS, providing a road map for future analyses.

## Methods

**Study approval**. This research was approved by the institutional research ethics boards (REB) at the Hospital for Sick Children (#1000046672), University of Toronto (#18-5553), and the Leeds East Local Research Ethics Committee (#10/H1306/70). All zebrafish experiments were performed in accordance with the policies and guidelines of the Canadian Council on Animal Care and an institutionally approved animal use protocol (#41617).

**Human subjects**. Three individuals (from two families) were identified from a Canadian cohort and two additional individuals were identified in a UK cohort. Patients in Canada were referred to the MH Investigation Unit (MHIU) at Toronto General Hospital, where susceptibility to MH is diagnosed based on the North American standardization protocol for Caffeine Halothane Contracture Test (CHCT) protocol[23]. Patients in UK were referred to the Malignant Hyperthermia Investigation Unit in Leeds after presenting clinically with EHI. In vitro contracture test (IVCT) was executed in accordance with the European Malignant Hyperthermia Group (EMHG) protocol[27], and the Heat Tolerance Assessment was undertaken as previously described[26]. Targeted next-generation sequencing was conducted for genetic testing as previously described[52].

**Whole-exome and whole-genome sequencing**. All genomic sequencing was performed at The Centre for Advanced Genomics (TCAG, SickKids, Toronto, Canada). Whole-exome and genome sequencing were performed as previously described at a minimum average read depth of 50x[53]. Sequences were aligned to GRCh37.p13 using bwa (v. 0.7.12) aligner and variant calling was performed using GATK3.7 following best practices. After variant calling variants were annotated using VEP (v. 94)[54]. Rare (MAF < 0.01) variants were identified through comparison with gnomAD and ExAC.

**ASPH transcripts for junctin**. The transcripts for skeletal muscle junctin in human and mouse were referenced to ENST00000522603 and ENSMUST00000103004, respectively, based on Ensembl genome browser.

**Establishing stable transgenic zebrafish**. Tol2 transposase-based system was applied for transgenesis as previously described[55,56]. Briefly, the mCherry coding sequence followed by human junctin coding sequence was cloned into the pDONR221 (Tol2Kit) and then cloned in a Tol2 transposon backbone using P5E-503unc (Addgene plasmid #64020), pME-mCherry-junctin, p3E-polyA (Tol2Kit) and pDESTtol2pA2 (Tol2Kit). The plasmids, pDESTtol2pA; 503unc:mCherry-junctin-polyA, were co-injected with Tol2 RNA into one-cell stage zebrafish (AB strain, ZIRC). Adult F0 fish were outcrossed to wild-type AB fish, and the founder embryos were then raised to adulthood (F1). To limit variability, F2 lines were selected whose RNA expression levels in mCherry-junctin were the closest to Tg-wild type (Tg-WT). The selected F2 lines and F3 progeny were used for this study.

**qRT-PCR analysis**. Five F2 embryos were gathered in one sample and total RNA was isolated using the RNeasy mini kit (QIAGEN), and 500 ng was reverse-transcribed using the iScript™ cDNA Synthesis Kit (Bio-Rad). Quantitative PCR was performed using SYBR Green (Thermo Fisher) with each reaction contained cDNA derived from 10 ng total RNA, and the Applied Biosystems StepOne Real-Time PCR System (Thermo Fisher Scientific). mRNA levels of mCherry-junctin were analyzed using the 2-ΔΔCT method as previously described[57] with eef1a1l1 as an internal control. Three technical replicates of each sample and four samples from each group were tested. Primers used were: 5′-TGATGTGGATGATGCCAAAG-3′ (junctin forward), 5′-CAGGTTTCTCTTTCTCCTTCTTG-3′ (junctin reverse), 5′-CTGTTACCTGGCAAAGGGGA-3′ (eef1a1l1 forward), 5′-CGTGGCCAATAACCACGATG-3′ (eef1a1l1 reverse).

**RNA injection to create transient human junctin-expressing zebrafish**. To generate templates for RNA synthesis, mCherry coding sequence followed by human junctin coding sequence including wild-type, V54A (c.161T > C), K88T (c.263A > C), D149H (c.445G > C), and K202R (c.605A > G) was cloned into pCS2 + MT (provided by Dr. David L. Turner in University of Michigan). Capped transposase RNA was synthesized using mMESSAGE mMACHINE SP6 Transcription Kit (Invitrogen). 400 pg of RNA was injected into the one-cell stage zebrafish embryo yolk. Injected embryos were screened for mCherry fluorescence at 2 dpf.

**Fluorescence immunostaining of isolated skeletal myofibers**. The transgenic embryos were disassociated at 5 dpf, and isolated myofibers were plated on cover glass slips as previously described[58]. Myofibers were fixed with methanol and blocked with 10% goat serum and 1% BSA in PBST (0.1% Tween in PBS). Anti-mCherry antibody (1:500; Abcam ab167453) and anti-RYR antibody 34C (1:100; Abcam ab2868) were used for primary antibodies. Images were taken using the Leica SP8 Lightning Confocal microscope.

**Electron microscopy**. Anaesthetize fish larvae in 0.1% tricaine at 5 dpf were fixed in 2% paraformaldehyde and 2.5% glutaraldehyde in 0.1 M sodium cacodylate buffer. Post-fixation, embedding, sectioning, and staining were conducted as previously described[56]. Samples were imaged using an FEI Tecnai 20.

**Heat test and drug treatment on fish**. Each fish group was placed in a 6-well with 6 ml of autoclaved egg water. Thermal stimulation was provided by placing the plate on a heat block set at 34 °C or 37 °C. Caffeine (Sigma-Aldrich C0750) or 4-Chloro-3-methylphenol (4CmC, Sigma-Aldrich C55402) was added to the egg water final concentrations of 1 mM and 10 μM, respectively. Dantrolene (Sigma-Aldrich D9175) or N-Acetyl-L-cysteine (NAC, Sigma-Aldrich A7250) was added to the system water at concentrations of 5 μM and 200 μM, respectively, for 2 h immediately before thermal stimulation and was removed at the end of the thermal stimulus.

**Swimming assay**. To assess the motor function, 3-dpf fish were individually transferred to a 96-well plate and incubated with 10 μM optovin 6b8 (ChemDiv) at 28.5 °C for 5 min as previously described[59]. The embryos exposed to white light for 10 s to elicit swimming, repeated three times with an interval of 1 min, were monitored and analyzed using ZebraBox (ViewPoint). Three independent experiments were conducted, and total distance traveled (mm) during the 30-s of light exposure were plotted and analyzed using statistics software (GraphPad Prism 9).

**Caffeine dependence of Ca²⁺ release in zebrafish myofibres**. Zebrafish myofibers were dissociated with collagenase as previously described[58]. Myofibers were loaded with 5 μM fluo-4AM (Molecular Probes) for 45 min at room temperature in a normal rodent Ringer's solution consisting of (in mM): 145 NaCl, 5 KCl, 2 CaCl₂, 1 MgCl₂, 10 HEPES, pH 7.4, followed by de-esterification in normal rodent Ringer's solution supplemented with 25 μm N-benzyl p-toluene sulfonamide (BTS) for 20 min at room temperature to inhibit contractions. Fluo4-AM loaded myofibers were excited at 480 ± 15 nm and fluorescence emission detected at 535 ± 20 nm was collected at 10 kHz using a photomultiplier system. Caffeine responses were obtained by sequential exposure of fluo-4AM loaded myofibers to various concentrations of caffeine applied through a rapid (response time <5 s) local perfusion system (Warner Instrument Corporation, Hamden, CT). For these experiments, myofibers were exposed sequentially for 30 s to different concentrations of caffeine from 0.3 mM followed by 30 s with either 0.7 mM, 1.0 mM, 3 mM, 10 mM, or 30 mM caffeine with each concentration followed by a 30 s wash with control solution.

**Zebrafish 2′,7′-dichlorofluorescin diacetate (DCFDA) assay**. To detect intracellular oxidant activity, the generation of reactive oxygen species (ROS) in 6-dpf fish was measured using DCFDA (Sigma-Aldrich D6883) as previously described[60]. Briefly, 20 of euthanized larvae were homogenized in 100 μL cold buffer composed of 0.32 mM sucrose, 20 mM HEPES and 1 mM MgCl₂ (pH = 7.4), plus protease inhibitor (Roche 11836170001). The homogenate was centrifuged at 20,000 × g for 20 min at 4 °C, and 20 μL of supernatant was transferred to a 96-well plate with three technical triplicates. 100 μL of phosphate-buffered saline and 8.3 μL DCFDA stock solution (10 mg/mL, Sigma D6883) were added to each well and incubated at 37 °C for 30 min. The fluorescence intensity was measured using a microplate reader (Varioskan plate reader) with excitation and emission at 485 and 535 nm, respectively. Values were normalized to protein concentrations and showed in relative fluorescent units (RFU).

**Assessment of muscle damage in zebrafish**. To measure the severity of muscle damage after heat challenge, immunofluorescence was conducted using whole-mount zebrafish embryos as previously described[61]. 3-dpf embryos exposed to 1 mM caffeine at 34 °C for 1 h the day before were fixed in 4% paraformaldehyde in PBS. Anti-myosin antibody (1:10; DSHB A4.1025) was used for the primary antibody. Images were taken using the Leica SP8 Lightning Confocal microscope.

Para-sagittal section in the trunk region was observed at the depth where both fast and slow muscles could be viewed. The severity of the damage was scored by assessing four somites per sample: Score 0—all myofibers were aligned and attached to myoseptam; Score 1—a few superficial myofibers detached from myoseptum; Score 2—severely injured myotome (SIM) ≤ 25%; Score 3—SIM ≤ 50%; Score 4—SIM ≤ 75%; Score 5—75% < SIM. A myotome in which multiple deep muscles detached from the myotendinous junctions (MTJs) and formed debris is defined as a SIM in this study. Ten fish per group were assessed.

**C2C12 Cell culture and differentiation to myotubes**. C2C12 myoblasts (ATCC CRL1772) were grown in growth medium containing 20% fetal bovine serum (FBS) and antibiotic-antimycotic solution (Wisent Bioproducts 450-115-EL) in Dulbecco's modified Eagle's medium (DMEM) in a 5% CO₂ humidified atmosphere at 37 °C. For myotube experiments, cells were seeded on collagen (Thermo Fisher Scientific A1048301) coated plastic bottom dish or Matrigel Matrix (Corning) coated coverslips, and medium was changed to differentiation medium containing 2% fetal horse serum and antibiotic-antimycotic solution in DMEM once cells reached 80–90% confluency. The medium was changed every 48 h.

**Generating junctin mutant C2C12 cells using CRISPR-Cas9 strategy**. Double nicking with Cas9n and homology-directed repair (HDR) with single-stranded oligodeoxynucleotide (ssODN) were applied for generating junctin knock-in C2C12 cells[62]. The single-guide RNA (sgRNA) against Exon 2 and Exon 4 were 5′-GAAGTTCTAGGTAAGAACTA-3′ and 5′-GACCATGAACCATGTGAAA A-3′, 5′-GTTAGCCAAGAGGAAAACTA-3′ and 5′-GGTATTTACAAGATAA AAAG-3′, respectively (Supplementary Fig. 6). Each of the sgRNA sequences was cloned into pSpCas9n(BB)-2A-Puro (PX462) V2.0 (Addgene plasmid # 62987). The ssODNs consisted of 200 bases centered on the knock-in location were synthesized by Integrated DNA Technologies. The two plasmids of 1 μg each and 1.0 nmol of the ssODN were co-transfected into C2C12 cells by electroporation (Amaxa Nucleofector Lonza). At 72 h post-transfection, the cells were selected in media containing 4 mg/mL puromycin for three days and then subcloned into 96-well plates. Once at sufficient cell density, the genomic DNA of subclones was analyzed by Sanger sequencing to detect mutations. Primers used for Exon 2 and Exon 4 were (forward) 5′-ACCCAATCAGGAGTTTGCTT-3′ and (reverse) 5′-CA GCCCTTTGCTTTTTCAAG-3′, and (forward) 5′-TGGTGCTGACTGGTTCTC AG-3′ and (reverse) 5′-GCATGTCTTCCATGCATCC-3′, respectively. Double-stranded DNA breaks with Cas9 and nonhomologous end-joining (NHEJ) were induced to generate junctin knock-out C2C12 cells[63]. The sgRNA against the transmembrane region in Exon 2 was 5′-GGACATCTGTGGCTGTCGTG-3′ and cloned at the BbsI site into pSpCas9(BB)-2A-Puro (PX459) V2.0 (Addgene plasmid # 62988). The plasmid of 2 μg was transfected into C2C12, and the rest of the process was the same as above.

**Western blotting of cells**. C2C12 myotubes 7 days after starting differentiation were lysed with cell lysis buffer (Cell Signaling #9803) containing protease inhibitor (Roche 11836170001) and phosphatase inhibitor (Millipore 524625). Aliquots containing 25 μg of protein were resolved by SDS-PAGE, and transferred onto a PVDF membrane (Bio-Rad #1620174) according to standard procedures. Total protein stains were conducted using Rever 700 Total Protein Stain (LI-COR Biosciences) followed by imaging with the Odyssey Fc Imager (LI-COR). Primary antibodies used were junctin (1:5000, gifted by Dr. Angela Dulhunty), CASQ1 (1:2000, ab191564, Abcam), RyR (1:200, 34 C, DSHB), and myosin (1:200, A4.1025, DSHB). Anti-rabbit or anti-mouse IgG-HRP conjugate (1:5000, Bio-Rad) were used for secondary antibody.

**Caffeine dependence of Ca²⁺ release in C2C12 myotubes**. C2C12 myotubes were cultured onto white wall, clear-bottom 96-well plates (Costar #3610) until 6 days after starting differentiation. Cells were washed with 1x regular rodent Ringer's solution (RRS: 145 mM NaCl, 5 mM KCl, 2 mM CaCl₂,1 mM MgCl₂, 10 mM Hepes, pH 7.4) and then loaded with 5 μM Fur-2 AM in RRS for 1 h at 37 °C. Dye was then replaced with 90 μl of RRS. Using the on-board pipettor feature of the Flexstation3 (Molecular Devices), 10 μl of various concentrations of 10X agonist were added at 30 s and the whole protocol ran for a total of 200 s either at room temperature or 37 °C. The fluorescence signal was measured for 200 s and we analyzed data as peak 340 nm/380 nm ratio.

**Measurement of intracellular calcium in C2C12 myotubes**. Intracellular calcium [Ca²⁺]i was measured in day-6 myotubes as previously described[64]. Myotubes on coverslips were loaded with 5 mM Fura-2 AM (Invitrogen) in 1x regular rodent Ringer's solution (RRS) at RT for 45 min. For measurement in heat and caffeine status, myotubes were incubated with 50 μM 4CmC at 37 °C for 10 min just after loaded Fura-2 AM, and then mounted in a chamber set at 37 °C on the stage of an epifluorescence-equipped inverted microscope (Zeiss). Myotubes were sequentially excited at 340 and 380 nm wavelength and fluorescence emission at 510 nm was collected using a high-speed CCD camera (Hamamatsu). The ratio of the fluorescence intensity (R340/380) was analyzed using Volocity Software (PerkinElmer). For calculating the intracellular calcium concentration, the maximum ratio (Rmax) and minimum ratio (Rmin) were determined by using ionomycin (20 μM) in RRS

followed by EGTA solution (125 mM) at the end of each experiment. [Ca²⁺]i of each myotube was calculated as previously described[65].

**Reactive oxygen species (ROS) in C2C12 cells**. All wells on 96-well plate were coated with Matrigel Matrix (Corning) and 5000 myoblasts were seeded per well a day before starting differentiation. Day-6 myotubes were assessed for ROS production. For thermal stimulation, plates were taken out from an incubator and placed at RT for 1 h. Myotubes were washed with PBS and added 100 μL RRS per well, and then scanned to measure the background intensity (F_bg) using a microplate reader (Varioskan plate reader) with excitation at 475 nm and emission at 535 nm. After wash with PBS, myotubes were incubated with 50 μM DCFDA (Sigma-Aldrich D6883) at RT for 30 min. After removal of DCFDA solution, 100 μL RRS was added into each well and the plates were scanned to measure the fluorescence intensity (F_0min). 50 μL of RRS was removed from each well and 50 μL of the solution was added instead to achieve 1 mM caffeine or 50 μM 4CmC. The plates were warmed in an incubator at 37 °C for 30 min, and then scanned to capture the fluorescence intensity (F_30min). Each condition was tested with at least eight technical replicates, and three independent experiments were repeated. The data are expressed as fold change (F_30min – F_bg / F_0min – F_bg).

**Immunoprecipitation of RyR1 and analysis of phosphorylation**. WT and junctin mutant C2C12 myotubes (day 6, $n = 4$ independent replicates of WT) in 50 mM Tris-HCl, pH 7.5, 170 mM NaCl, 1 mM EDTA, and 0.5% NP-40 containing protease/phosphatase inhibitors (IP buffer) were homogenized with a bead homogenizer (Precellys 24, Bertin Technologies, 25 s 4 cycles), and 10 mg of the cell lysates were used for immunoprecipitation. To remove non-specific binding proteins (beads control), 30 μl of magnetic beads (Dynabeads™ Protein G, Invitrogen) was added to cell lysates and incubated at 4 °C for 30 min. The beads were separated by pacing tubes on magnetic stand. The supernatants were transferred to new tubes and RyR antibody (5 μg, 34C, DSHB) was added. After 1 h of incubation with RyR antibody at 4 °C, magnetic beads (30 μl) were added and incubated for 30 min. Beads were washed three times with IP buffer. The affinity-purified RyR1 and the bead control were placed in 20 μl of 20 mM Tris (pH 8.0). Proteins were first alkylated by adding DTT (1 μl of 100 mM) was added and incubated for 30 min with agitation. Following the incubation, then acryl amide (1.2 μl of 100 mM) was added and incubated for 10 min. 500 ng of MS- grade trypsin/lys-c protease mix (A40007, Thermo Scientific) in 10 μl of 20 mM Tris (pH 8.0) was added to the beads and digested for overnight at 37 °C. The supernatant was removed, transferred to a new tube, and further digested with 500 ng of trypsin (T9600, Gendepot) for 3 h. The digested peptide was enriched by in-housed STAGE tip column with 2 mg of C18 beads (3 μm, Dr. Maisch GmbH, Germany) and vacuum dried. Resuspended peptides were subjected to a nanoLC-1000 (Thermo Scientific) coupled to Orbitrap Fusion mass spectrometer (Thermo Scientific) with ESI source. The peptides were loaded onto an in-house Reprosil-Pur Basic C18 (1.9 μm, Dr. Maisch GmbH, Germany) trap column (2 cm length, 100 μm i.d.) and separated by 5 cm column (150 μm i.d.) with a 90 min gradient of 2–28% of acetonitrile/0.1% formic acid at a flow rate of 950 nl/min. The data acquisition was made in data-dependent analysis mode (DDA) for unbiased peptide detection and targeted parallel reaction monitoring mode (PRM) for deep monitoring of phosphorylated RyR1 S2844 residue. For DDA mode, precursor MS spectrum was scanned at 300–1400 $m/z$, 120k resolution at 400 $m/z$, $5 \times 10^5$ AGC target (50 ms maximum injection time) by Orbitrap. Top 1 s cycle time was applied to selected MS1 signal and filtered by Quadrupole (2 $m/z$ isolation window, 20 s exclusion time), fragmented by collision-induced dissociation (CID), and detected by Ion trap with rapid scan rate ($5 \times 10^3$ AGC target, and 35 ms of maximum injection time). For PRM mode to detect RyR1 S2844 phosphorylated peptide the target $m/z$ was 680.3007 (+2 charge, I[p]SQTAQTYDPR), 744.3481 (+2 charge, KI[p]SQTAQTYDPR), 469.5679 (+3 charge, KI[p]SQTAQTYDPR). Pre-selected precursor ions were scanned for 0 to 90 min of MS run time and isolated by quadrupole followed by collision-induced dissociation (CID) MS2 analysis. Obtained spectra were searched against the target-decoy mouse RefSeq database (release 2020) in Proteome Discoverer 2.1 interface (PD 2.1, Thermo Fisher) with the Mascot algorithm (Mascot 2.4, Matrix Science). Dynamic modifications of the phosphorylation on serine, tyrosine and threonine, acetylation of N-terminus and oxidation of methionine were allowed. The precursor mass tolerance was confined within 20 ppm with fragment mass tolerance of 0.5 Da and a maximum of two missed cleavages was allowed. Assigned peptides were filtered with 1% false discovery rate (FDR) using percolator validation based on $q$-value then imported to Skyline with raw data file. We validated each result by deleting non-identified spectrum and adjusting the area-under-curve (AUC) range. The percentage of S2844 residue phosphorylation was shown as [sum of AUC of phosphorylated S2844 containing peptides]/[sum of AUC of phosphorylated S2844 containing peptides + sum of AUC of non-phosphorylated S2844 containing peptides].

**Affinity purification-mass spectrometry (AP-MS)**. To prepare antibody-conjugated magnetic beads, 25 μL of Dynabeads Protein A (Life Technologies) were incubated with anti-mCherry antibody (Abcam ab167453) for 1 h at RT. Beads and antibodies were covalently crosslinked using BS3 Crosslinker (ThermoFisher).

Sixty fish at 6 dpf were homogenized in lysis buffer (150 mM NaCl, 50 mM Tris-HCl, 1 mM EDTA, 1% CHAPS, pH 8.0) supplemented with 1 mM PMSF, 1 mM TCEP, and protease inhibitor protease inhibitors. The homogenate was incubated on ice for 30 min, and then centrifuged at $20,000 \times g$ for 20 min at 4 °C. The supernatant was transferred to a tube containing pre-conjugated beads and incubated for 30 min at 4 °C. Beads were washed once in lysis buffer, and twice in wash buffer (20 mM Tris-HCl, 150 mM NaCl, 0.3% CHAPS, pH 8.0). Samples were then trypsin digested on beads as previously described[66], and digested peptides were analyzed using a nano-HPLC (High-performance liquid chromatography) coupled to MS. One-quarter of the sample was used. Nano-spray emitters were generated from fused silica capillary tubing, with 100 μm internal diameter, 365 μm outer diameter, and 5–8 μm tip opening, using a laser puller (Sutter Instrument Co., model P-2000, with parameters set as heat: 280, FIL = 0, VEL = 18, DEL = 2000). Nano-spray emitters were packed with C18 reversed-phase material (Reprosil-Pur 120 C18-AQ, 3 μm) resuspended in methanol using a pressure injection cell. Sample in 5% formic acid was directly loaded at 800 nl/min for 20 min onto a 100 μm × 15 cm nano-spray emitter. Peptides were eluted from the column with an acetonitrile gradient generated by an Eksigent ekspert™ nanoLC 425, and analyzed on a TripleTOF™ 6600 instrument (AB SCIEX, Concord, Ontario, Canada). The gradient was delivered at 400 nl/min from 2% acetonitrile with 0.1% formic acid to 35% acetonitrile with 0.1% formic acid using a linear gradient of 90 min. This was followed by a 15 min wash with 80% acetonitrile with 0.1% formic acid, and equilibration for another 15 min to 2% acetonitrile with 0.1% formic acid. The total DDA protocol is 135 min. The first DDA scan had an accumulation time of 250 ms within a mass range of 400–1800 Da. This was followed by 10 MS/MS scans of the top 10 peptides identified in the first DDA scan, with accumulation time of 100 ms for each MS/MS scan. Each candidate ion was required to have a charge state from 2 to 5 and a minimum threshold of 300 counts per second, isolated using a window of 50mDa. Previously analyzed candidate ions were dynamically excluded for 7 s. Mass spectrometry data generated were stored, searched, and analyzed using ProHits laboratory information management system (LIMS) platform. Within ProHits, WIFF files were converted to an MGF format using the WIFF2MGF converter and to an mzML format using ProteoWizard (V3.0.10702) and the AB SCIEX MS Data Converter (V1.3 beta). The data was then searched using Mascot (V2.3.02) and Comet (V2016.01 rev.2). The spectra were searched with the Danio rerio sequences (version 65) and Homo sapiens sequences (V57) in the RefSeq database acquired from NCBI, supplemented with "common contaminants" from the Max Planck Institute (http://maxquant.org/contaminants.zip) and the Global Proteome Machine (GPM; ftp://ftp.thegpm.org/fasta/cRAP/crap.fasta) and forward and reverse sequences (labeled "gi|9999" or "DECOY") for a total of 158,998 entries. Database parameters were set to search for tryptic cleavages, allowing up to 2 missed cleavages sites per peptide with a mass tolerance of 35ppm for precursors with charges of 2+ to 4+ and a tolerance of 0.15 amu for fragment ions. Variable modifications were selected for deamidated asparagine and glutamine and oxidized methionine. Results from each search engine were analyzed through TPP (the Trans-Proteomic Pipeline, v.4.7 POLAR VORTEX rev 1) via the iProphet pipeline. Mass spectrometry analysis was performed by the Network Biology Collaborative Centre (NBCC) at Lunenfeld-Tanenbaum Research Institute (Toronto, ON).

**Interaction data analysis and visualization using ProHits-viz**. Significance analysis of interactome (SAINT) express version 3.6.3 was used to score the enrichment of proteins in samples compared to negative controls using default parameters[67]. Two biological replicates of mCherry-empty samples were used as negative controls. Proteins with a Bayesian (B)FDR ≤ 1% and at least two unique peptides were considered true positive interactors. The entire network, scored with SAINTexpress, was used to generate input files for ProHits-viz[68] for dot plot generation. To be included in a dot plot, a prey had to pass the 1% BFDR threshold with at least one bait. Once included, the prey's quantitative values were recovered across all baits and methods, regardless of the BFDR. Unless otherwise indicated, the default options were selected, including "control subtraction" and hierarchical clustering using the Canberra distance metric and the Ward clustering type. Average spectral count values were capped at the maximum indicated in each figure, and unless otherwise indicated, no minimum spectral counts were required and no additional normalization steps were performed.

## Data availability

The identified *ASPH* variants have been deposited to the ClinVar database (https://www.ncbi.nlm.nih.gov/clinvar/). Whole-exome and whole-genome sequencing data will not be available to the public because of the potential for compromising the privacy and consent of study participants. Information on the sequence raw data supporting the results of this study is available from the corresponding authors upon request. Proteomics data have been deposited to the ProteomeXchange Consortium (http://www.proteomexchange.org/) via the MassIVE repository (https://massive.ucsd.edu/ProteoSAFe/static/massive.jsp). ProteomeXchange identifiers are PXD033745 (dataset for Fig. 4e) and PXD033578 (dataset for Fig. 4f, Supplementary Figs. 9 and 10). MassIVE repository accession IDs are MSV000089430 (Fig. 4e) and MSV000089417 (Fig. 4f, Supplementary Figs. 9 and 10). The analyzed data presented in this paper are provided in the Source data file. Source data are provided with this paper.

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

## Acknowledgements

We thank the patients and families for their participation in this study. Primary funding support was through the Canadian Institutes of Health Research (CIHR 148603 to J.J.D.). Additional grant support was from the National Institutes of Health (R01AR072602 to S.L.H. and R01AR078000 to J.J.D./R.T.D.).

## Author contributions

J.J.D. and S.R. conceived the project. J.J.D. and Y.E. designed and conducted the majority of the experiments and interpreted the data. J.J.D. and Y.E. wrote the manuscript. R.T.D., S.R., P.M.H., and S.L.H. edited and corrected the manuscript. N.K., L.G., M.S., P.M.H., and S.R. provided resources and clinical information and conducted the genomic analysis. A.C. contributed to the bioinformatics analysis of WES and WGS. L.G. and R.T.D. performed assays for caffeine-induced SR calcium release in zebrafish myofibers and C2C12 myotubes. C.S.L., S.Y.J., and S.L.H. analyzed RyR1 phosphorylation.

## Competing interests

The authors declare no competing interests.
