## [Transparent Peer Review File · Nature Communications]

Peer Review Information

Manuscript title: Variants in ASPH cause exertional heat illness and are associated with malignant hyperthermia susceptibility

Corresponding author name(s): James Dowling

Reviewer comments & decisions:

Reviewer comments, first version:

Reviewer #1 (Remarks to the Author: Overall significance):

This manuscript identified a candidate gene for exertional heat illness/malignant hyperthermia from two different patient cohorts. All variations in ASPH were heterozygous, suggesting a dominant affect. This and the potential impact of these variants on swimming behavior with/without heat and caffeine challenge were next investigated. The experiments using transgenic/RNA – expressing zebrafish are well done and clearly show that pathogenetic variants in ASPH impact swimming behavior and muscle structure in the zebrafish model. The mechanism may be partially through excess ROS, as an anti-oxidant blunts the damage. Similar experiments in C2C12 cells led to similar, though less striking, results. Biochemical experiments suggest that Junctin interactions with RyR and CASQ1 may mediate Junctin's impacts on calcium homeostasis, muscle structure, and function. The impact of this manuscript is elucidating new genetic causes of EHI/MH, and identifying that the zebrafish model recapitulates important aspects of the phenotype and shows similar drug responsiveness – thus this is an excellent vertebrate model for these important conditions. Signed: Clarissa Henry

Reviewer #1 (Remarks to the Author: Strength of the claims):

The data are rigorous and well described. The authors very clearly state in which experiments they used established transgenic lines and which involved mRNA injections - showing similar results from both types of experiments bolsters their data. I only had one "major" comment - although not particularly major.

Major:

1. The variation in swimming distance is somewhat high – with controls swimming different distances (up to 25% difference) in different experiments. This is not alarming, the authors are to be lauded for showing their data clearly, but does raise the question of whether it makes sense to normalize data compared to controls on any given day in a given experiment?

Minor:

1. First bit of results section – relies on reader having read introduction but it's potentially worth quickly reiterating that excitation-contraction coupling (ECC) genes are strong candidates....
2. "We performed general characterization of the transgenic zebrafish lines, and identified no obvious differences in baseline motor behavior, muscle ultrastructure, or survival (Figure 2b)." Should change to Figure 2b-d
3. Add grant number "Primary funding support was through the Canadian Institutes of Health Research (CIHR # to JJD)"
4. What does dantrolene do and how does that impact thinking about mechanisms?

Reviewer #1 (Remarks to the Author: Reproducibility):

Manuscript is well written and described.

Reviewer #2 (Remarks to the Author: Overall significance):

In the manuscript entitled "Variants in ASPH cause exertional heat illness and are associated with malignant hyperthermia susceptibility.", by Endo et al, the Authors described a novel function of junctin, a variant of the aspartate beta-hydroxylase (ASPH) gene, on the treatment of exertional heat illness (EHI), and malignant hyperthermia susceptibility (MHS) muscle diseases. In the well-written and clear manuscript, the Authors showed that specific mutations found in human muscle samples on junctin (K88T and V54A) are relevant to keep the stability of ryanodine receptors during the excitation-contraction coupling, especially during high temperature and under the effect of halothane. The interesting approach taken by the Authors to understand these mutations was to transfer the mutated gene into two different and wide-used pre-clinical muscle models: zebrafish and C2C12 cell line. The findings in this manuscript are clinically relevant for the EHI and MHS patients, and for finding new alternative approaches for those diseases. There are nevertheless, minor comments that the Authors should address before a final decision.

- 1) The Authors should give the (in extenso) name of the gene abbreviation.
- 2) At the end of the introduction, the Authors only mentioned junctin as one candidate for EHI/ MHS,

without describing how this gene was selected, the function of the protein, and to whom it interacts. These pieces of information are necessary to understand the reason for the work, especially for those readers who are not familiar with the topic. Part of the explanation was found in the results, but still should be present in the introduction for better comprehension of the manuscript.

3) From the introduction, it is difficult to clearly see the aim of the work. The Authors should elaborate a sentence defining the aim(s) and the pieces of evidence to support the aim(s).

4) The Authors should describe in the results the concentration of the drugs used in the study. Moreover, it is lacking on figures 2 and 3 a description of the statistical analyses of whether the pharmacological treatment did or did not produce a significant result different from the control condition.

5) In-text for figure 2, the Authors state “, and identified no obvious differences in baseline motor behavior, muscle ultrastructure, or survival (Figure 2B).” Although there are no differences, the Authors only showed muscle ultrastructure data, without the other parameters. Maybe just a re-adjust of the position of the label (Figure 2B) will correct the imprecision of the text.

6) The Authors should clarify in the results whether the experiments using C2C12 cells were done in myoblast or myotubes, and if so, the passage, confluency, and days after differentiation.

7) In Figure 4e, the truncated data is represented with a red bar, instead of the orange bar from the other experiments. Is there a specific reason for the change of color?

8) In the paragraph of the discussion, starting with “Our data expand the knowledge... and positive CHCT in his sibling” the Authors should provide the missing citations.

Reviewer #2 (Remarks to the Author: Impact):

The current and the following works on the function of junctin will provide important and relevant pre-clinical and clinical information on the new therapies for EHI and MHS.

Reviewer #2 (Remarks to the Author: Strength of the claims):

The data presented in the current manuscript are enough to convince for further studies of junctin related to EHI nad MHS.

Reviewer #2 (Remarks to the Author: Reproducibility):

The Authors took care of the statistical analyses, giving enough information for the continuity work from independent groups.

Reviewer #3 (Remarks to the Author: Overall significance):

In this manuscript, the authors integrated human genetics, zebrafish genetics and cell culture models to discover causative genes for exertional heat illness (EHI) and malignant hyperthermia (MH). They performed genomic sequencing on a cohort with EHI/MH and identified rare, pathogenic variants in ASPH. They then generated transgenic zebrafish for two of the variants, which recapitulated the corresponding phenotypes in human. They went on to generate knock in alleles in C2C12 cell and also obtained promising data to prove causality. The logical flow is clear, and data and their conclusions are convincing, and the presentation is excellent.

Reviewer #3 (Remarks to the Author: Impact):

Successful modeling of the disease in the efficient zebrafish model is interesting, which shall have great potential for testing additional variants that will be identified in the future. Thus, the impact could be high.

Reviewer #3 (Remarks to the Author: Strength of the claims):

I do have the following concerns.

1. As to the transgenic fish, please comment on the promoter that you used. Whether the promoter drives gene expression in somites or ubiquitously in the whole body?
2. Page 6, the end of the second paragraph: WT and K88T junction mRNA levels were comparable, V54A mRNA was reduced. Please provide some explanation on this phenomenon?
3. Fig. 3A and 4B. Are the units the same? If so, please choose one. It seems data in fish and cell are different. Why?

Reviewer #3 (Remarks to the Author: Reproducibility):

Statistical analysis is adequate.

Author rebuttal, first version:

Response to Reviewers

We thank the three reviewers for their overall favorable evaluation of our manuscript. We appreciate the suggestions and comments, and have made all attempts to address the points raised by each reviewer. Please see point-by-point responses to each critique.

Reviewer #1

We thank reviewer 1 for overall positive impression and for the helpful comments.

1. The variation in swimming distance is somewhat high – with controls swimming different distances (up to 25% difference) in different experiments. This is not alarming, the authors are to be lauded for showing their data clearly, but does raise the question of whether it makes sense to normalize data compared to controls on any given day in a given experiment?

We appreciate this excellent point, which acknowledges what reviewer 1 well knows, that there can be clutch-to-clutch variability (even amongst controls) in terms of swim behavior. We agree with the suggestion to normalize the data to controls from the same day and experiment, and have now done this in two ways. The first was to normalize the means of each replicate to the means of the controls (non transgenic for transgenic lines and non injected for the RNA injected studies); the second was to normalize each data point (i.e. each fish studied) to the mean of the controls measured on the same day. The overall conclusion from this is that the data remain significantly different and with a high magnitude of difference. We now present in Figure 2d, 2e and 3c the re-evaluated data normalized for the individual fish, and present the normalization of the means in Supplemental Figure 4. Of note, we performed two additional replicates of the “heat plus caffeine” condition, as we viewed this as the most important condition in terms of variant validation. We did these replicates to provide additional confirmation of the significance and magnitude of change when normalizing the data to controls of the same day/experiment.

1. First bit of results section – relies on reader having read introduction but it’s potentially worth quickly reiterating that excitation-contraction coupling (ECC) genes are strong candidates....

We have added a sentence to the results section (line 5, page 5) to address this.

“Based on the known pathomechanisms underlying EHI/MH, variants in genes encoding components of the excitation contraction coupling machinery are strong causal candidates.”

2. “We performed general characterization of the transgenic zebrafish lines, and identified no obvious differences in baseline motor behavior, muscle ultrastructure, or survival (Figure 2b).” Should change to Figure 2b-d

We have corrected this as follows. “differences in baseline motor behavior (Figure 2d), muscle ultrastructure (Figure 2b), and survival (data not shown).”

3. Add grant number “Primary funding support was through the Canadian Institutes of Health Research (CIHR # to JJD)”

The grant number has been added.

4. What does dantrolene do and how does that impact thinking about mechanisms?

Dantrolene binds to RYR1 and reduces stimulated calcium release from the SR. Its effectiveness in our setting adds additional support to a causal link between *junctin* variants and MH/EHI. It also suggests that *junctin* variants “promote” MH/EHI via dysregulated calcium release through RYR1. We have added

additional wording to the discussion section to clarify dantrolene's mode of action and how it impacts consideration of junctin variant pathomechanisms (pg 13, the last paragraph).

Reviewer #2

We thank reviewer 2 for the positive comments and the helpful critiques.

1) The Authors should give the (in extenso) name of the gene abbreviation.

This has been added (pg 4, paragraph 2, line 2)

2) At the end of the introduction, the Authors only mentioned junctin as one candidate for EHI/ MHS, without describing how this gene was selected, the function of the protein, and to whom it interacts. These pieces of information are necessary to understand the reason for the work, especially for those readers who are not familiar with the topic. Part of the explanation was found in the results, but still should be present in the introduction for better comprehension of the manuscript.

We have added a new sentence to the end of the introduction. "Junctin is localized to the lumen of the terminal sarcoplasmic reticulum, where it has been shown to interact with calsequestrin and RYR1 and participate in the regulation of SR calcium dynamics²²."

3) From the introduction, it is difficult to clearly see the aim of the work. The Authors should elaborate a sentence defining the aim(s) and the pieces of evidence to support the aim(s).

We appreciate this suggestion, and have added the following to the last paragraph of the introduction. "The overall goal of this study is to identify new genetic causes of EHI/MHS."

4) The Authors should describe in the results the concentration of the drugs used in the study. Moreover, it is lacking on figures 2 and 3 a description of the statistical analyses of whether the pharmacological treatment did or did not produce a significant result different from the control condition.

We have added drug concentrations to the results section and have added a description of statistical methodology to these Figure legends. We also now compare treated junctin K88T mutants with matched controls, and show that treatment with either dantrolene or NAC restores mutant motor function (in the setting of heat+caffeine) to wild type levels (Figure 3C).

5) In-text for figure 2, the Authors state “, and identified no obvious differences in baseline motor behavior, muscle ultrastructure, or survival (Figure 2B).” Although there are no differences, the Authors only showed muscle ultrastructure data, without the other parameters. Maybe just a re-adjust of the position of the label (Figure 2B) will correct the imprecision of the text.

We have addressed this by changing the text to say the following: "differences in baseline motor behavior (Figure 2d), muscle ultrastructure (Figure 2b), or survival (data not shown)."

6) The Authors should clarify in the results whether the experiments using C2C12 cells were done in myoblast or myotubes, and if so, the passage, confluency, and days after differentiation.

Experiments were done on myotubes. The important parameters related to C2C12 handling have now been added to the methods section. In *Western blotting of cells* section (pg 21), C2C12 myotubes 7 days after starting differentiation; in *Caffeine dependence of Ca²⁺ release in C2C12 myotubes* section (pg 22), 6 days after starting differentiation; in *Measurement of intracellular calcium in C2C12 myotubes* section (pg 22), day-6 myotubes; in *Reactive oxygen species (ROS) in C2C12 cells* section. Day-6 myotubes; in *Immunoprecipitation of RyR1 and analysis of phosphorylation* section (pg 23), C2C12 myotubes (day 6).

7) *In Figure 4e, the truncated data is represented with a red bar, instead of the orange bar from the other experiments. Is there a specific reason for the change of color?*

We have corrected this so that the colors are consistent.

8) *In the paragraph of the discussion, starting with “Our data expand the knowledge... and positive CHCT in his sibling” the Authors should provide the missing citations.*

The appropriate citations have been added.

Reviewer #3

We thank reviewer 3 for the careful consideration and favorable evaluation of our manuscript.

1. *As to the transgenic fish, please comment on the promoter that you used. Whether the promoter drives gene expression in somites or ubiquitously in the whole body?*

The transgene is driven by the 503unc promoter, which results in muscle-specific expression. We have added the information to the results section (line 18, page 6).

2. *Page 6, the end of the second paragraph: WT and K88T junction mRNA levels were comparable, V54A mRNA was reduced. Please provide some explanation on this phenomenon?*

We do not have a precise explanation for this observation. The construct has been sequenced to confirm lack of presence of any second site mutations that may impact RNA stability. We can reflect that we have seen this occasionally with different RNAs, though this of course does not explain why the phenomenon occurs. However, we are confident in terms of the results generated with this construct, as even at lower RNA expression it still leads to abnormal swim behavior as compared to WT in response to heat plus caffeine.

3. *Fig. 3A and 4B. Are the units the same? If so, please choose one. It seems data in fish and cell are different. Why?*

These data were generated using two different techniques for evaluating calcium dynamics. For zebrafish single fiber experiments, we used a single wavelength dye (fluo-4), and the data (DF/F) is thus dimensionless. For the C2C12 cells, data are expressed as DRatio (also dimensionless), as these studies used fura-2 in Flexstation experiments.

Reviewer comments, second version:

REVIEWERS' COMMENTS:

Reviewer #1 (Remarks to the Author: Overall significance):

The authors have identified a new genetic cause of Exertional heat illness/Malignant hyperthermia - pathogenetic heterozygous variants in a gene that regulates excitation-contraction coupling. The authors rigorously validated their genomic sequencing data with C2C12 myotubes and transgenic zebrafish. Given the life threatening nature of EHI/MH, this is important new knowledge.

Reviewer #1 (Remarks to the Author: Strength of the claims):

Well done manuscript

Reviewer #1 (Remarks to the Author: Reproducibility):

Strong and rigorous

Reviewer #2 (Remarks to the Author: Overall significance):

In the revised manuscript entitled “Variants in ASPH cause exertional heat illness and are associated with malignant hyperthermia susceptibility.”, by Endo et al, the Authors described a novel function of junctin, a variant of the aspartate beta-hydroxylase (ASPH) gene, on the treatment of exertional heat illness (EHI), and malignant hyperthermia susceptibility (MHS) muscle diseases. Authors identified new specific mutations in human muscle samples on junctin (K88T and V54A), which results are relevant to keeping the stability of ryanodine receptors during the excitation-contraction coupling, especially during high temperature and under the effect of halothane. The interesting approach taken by the Authors to understand these mutations was to transfer the mutated gene into two different and wide-used pre-clinical muscle models: zebrafish and C2C12 cell line. The findings in this manuscript are clinically relevant for the EHI and MHS patients, and for finding new alternative approaches for those diseases.

Reviewer #2 (Remarks to the Author: Strength of the claims):

After the revision process, the authors improved the quality of the manuscript. No further experiments or revision is necessary.

Reviewer #2 (Remarks to the Author: Reproducibility):

The description of the methods and the statistical analyses were correctly done, providing enough information for reproducing the data.

Reviewer #3 (Remarks to the Author: Strength of the claims):

The authors have addressed all my concerns.

Author rebuttal, second version:

Response to Reviewers

We thank the reviewers and the editor for the favorable impression of the manuscript. No additional suggestions or comments were presented by the reviewers, so no point-by-point response is needed. We have incorporated all of the requisite editorial changes (these are highlighted in the checklist).